# Data-driven organic solubility prediction at the limit of aleatoric uncertainty

Lucas Attia [1,2], Jackson W. Burns [1,2], Patrick S. Doyle [1] & William H. Green [1] ✉

Small molecule solubility is a critically important property which affects the efficiency, environmental impact, and phase behavior of synthetic processes. Experimental determination of solubility is a time- and resource-intensive process and existing methods for in silico estimation of solubility are limited by their generality, speed, and accuracy. This work presents two models derived from the FASTPROP and CHEMPROP architectures and trained on BigSolDB which are capable of predicting solubility at arbitrary temperatures for a wide range of small molecules in organic solvent. Both extrapolate to unseen solutes 2–3 times more accurately than the current state-of-the-art model and we demonstrate that they are approaching the aleatoric limit (0.5–1 log $S$) of available test data, suggesting that further improvements in prediction accuracy require more accurate datasets. The FASTPROP-derived model (called FASTSOLV) and the CHEMPROP-based model are open source, freely accessible via a Python package and web interface, highly reproducible, and up to 2 orders of magnitude faster than current alternatives.

The solubility of organic solids in various solvents is an essential molecular property that impacts the efficiency[1], environmental impact[2,3], and phase behavior[4] of synthetic processes. Solubility is crucial in wide-ranging chemical processes spanning length and time scales including crystallization and filtration[4], membrane-based chemical separations[5,6], pharmaceutical design and discovery[7], drug delivery and formulation[8], the environmental fate of per- and polyfluoroalkyl substances (PFAS)[9] and geological-scale dissolved organic carbon flux[10]. By convention, solubility $S$ in $mol\,L^{-1}$ is expressed as $\log_{10}S$ since values can range over several orders of magnitude. Experimental methods for determining solubility are notoriously time- and resource-intensive[11] and error prone, thus many published values are suspected of being highly inaccurate. The challenges of measuring solubility are especially painful in pharmaceutical development where organic solubility complicates synthesis and purification[12] and aqueous solubility limits in vivo efficacy[13]. Given that solubility as a function of temperature is often desired, experimental determination becomes even more onerous. For these reasons a priori estimation of $\log S$ has long been of immense interest to the chemical sciences.

Critically, the experimental error is typically systematic rather than random because organic molecules are often isolated as an amorphous solid, hydrate, polymorph, or impure cocrystal rather than the desired most-stable pure crystal, confounding accurate measurement[14]. The reported standard deviation in inter-laboratory measurements in log $S$ typically ranges between 0.5 and 0.7 log units for aqueous solubility. For example, Katritzky et al.[15] notably found the average inter-laboratory standard deviation of 411 compounds to be 0.58. Other reported average standard deviations in inter-laboratory measurements include 0.6-0.7[16], 0.62[17], and 0.60[18]. Furthermore, other work found that inter-laboratory solubility measurements for the same solution could range over 0.86 log units and in some cases could vary as widely as 1.56 log unit[19]. Andersson et al.[20] recruited 12 laboratories to measure solubility, standardizing materials and collection methods between laboratories, and found that differences in data analysis alone could result in variations as high as a standard deviation of 0.74 log unit between labs. In sum, a variability between a factor of 3 (0.5 log units) and 10 (1 log unit) in the measured solubility of the same solute in aqueous solutions at the same temperature between laboratories is not unusual.

[1]Department of Chemical Engineering, MIT, Cambridge, MA, USA. [2]These authors contributed equally: Lucas Attia, Jackson W. Burns. ✉e-mail: whgreen@mit.edu

Prediction methods have evolved from empirical group additivity correlations[21,22], to ab initio conductor-like screening model (COSMO) and its extension to realistic solvation (COSMO-RS)[23], to bespoke-solvent machine learning (ML) models with random forest regressors[24]. Direct ab initio calculation of organic crystals using quantum mechanics is now possible but too computationally expensive for routine or high-throughput calculations[25,26]. Given the specific importance of aqueous solubility in drug discovery, most work has focused on predicting aqueous solubility[27], and relatively fewer works have explored organic solvents, which are particularly crucial in synthetic processes. The aforementioned experimental variability limit of 0.5-1 log $S$ represents a bound on the performance of any data-driven prediction method on a given dataset, since it is impractical to remeasure a large number of solute-solvent solubility numbers with significantly better accuracy and precision. Currently, this variability limit has primarily been explored in the context of aqueous solubility in the literature.[16,17,28] However, there is little reason to believe that the experimental uncertainty should be any lower in organic solvents, and may actually be higher due to increased variability in the experimental methodologies used across laboratories[29]. This variability defines the aleatoric limit - the 'irreducible error' below which model performance improvements cannot be discerned.

State-of-the-art methods focus on applying deep learning to organic solubility prediction, including graph-based neural networks and descriptor-based models[24,29–31]. Existing models, however, suffer from a lack of generalizability for a variety of reasons. Boobier et al.[24] trained solvent-specific models on only commonly available solvents at room temperature due to a lack of sufficient data to do otherwise, thus rendering their models non-generalizable by construction. Other works like those of Lee et al.[30] fail to evaluate model performance when extrapolating to new, unseen solutes, a task which mirrors the real task where solubility prediction would be applied in a synthetic pipeline. The state-of-the-art model in literature by Vermeire et al.[31] overcame some of these limitations by training a composition of deep learning models on the Gibbs free energy, enthalpy of solvation, and the Abraham solvation parameters, which are then combined via a thermodynamic cycle to predict the solubility in arbitrary solvents for a wide range of temperatures. A significant advantage of the Vermeire et al.[31] model is that when interpolating from known experimental data for a given solute to a new solvent the model predictions are very accurate. Unfortunately without experimental data to supplement the models, as would be the case when screening new solutes in a discovery pipeline, performance drops substantially.

Here, we combine advances in cheminformatics software and a recently compiled database of organic solubility, BigSolDB,[32] develop a state-of-the-art general organic solubility prediction model, and

validate it under rigorous extrapolation conditions. By adapting the FASTPROP[33] and CHEMPROP[34,35] architecures to ingest two molecular structures and a temperature (Fig. 1a) we train models on BigSolDB to regress log $S$ directly (Fig. 1b). Our optimized models yield a factor of three improvement over the existing state-of-the-art organic solubility prediction models, with rapid inference times suitable for use in high-throughput workflows. We further demonstrate that our optimized FASTPROP and CHEMPROP models, which use fundamentally different molecular representations, both reach the irreducible error, or aleatoric limit, of accuracy with only a small fraction of the total available data. Further progress in organic solubility prediction will require higher quality datasets to determine the true accuracy of the best models. Our fastest model, termed FASTSOLV, is open source and can be downloaded as a python package (pypi.org/project/fastsolv/), accessed online via fastsolv.mit.edu, and is integrated in ASKCOS (askcos.mit.edu).

## Results

### Datasets and model training approach

In a real discovery context, solubility prediction is usually applied toward a solute extrapolation task, wherein it is desirable to know the solubility of a novel candidate compound in a variety of standard solvents for a given temperature. Thus, we rigorously trained and evaluated our model performance with this task in mind. However, attention to the ability of a model to extrapolate to new solutes is not typically heeded in the literature, making benchmarking our model performance challenging.

We selected Vermeire et al.[31] as the current literature state-of-the-art model against which to benchmark our results, given that it is widely regarded as a highly accurate solvent and temperature-general model[36,37]. This approach consists of up to four machine learning sub-models trained on thermochemical datasets which are combined to yield solubility using a thermodynamic cycle. The overall model is tested on the experimental solubility data compiled in the SolProp dataset. Their objective was to to achieve excellent performance when extrapolating into new solvents, so the compiled SolProp dataset contains many solute structures already present in the training data (Fig. 2a). Since its publication, improved models for some constituent properties have become available, such as the solvation energy predictor developed by Kim et al.[37], but we leave the original SolProp model intact for the sake of fair comparisons with its originally published performance.

We trained our models on the BigSolDB dataset[32], which contains variable organic solvent and variable temperature solubility data at the precipitation limit. As a point of comparison with the model developed by Vermeire et al.[31], we tested our models on the SolProp dataset, though with overlapping solute structures dropped from BigSolDB

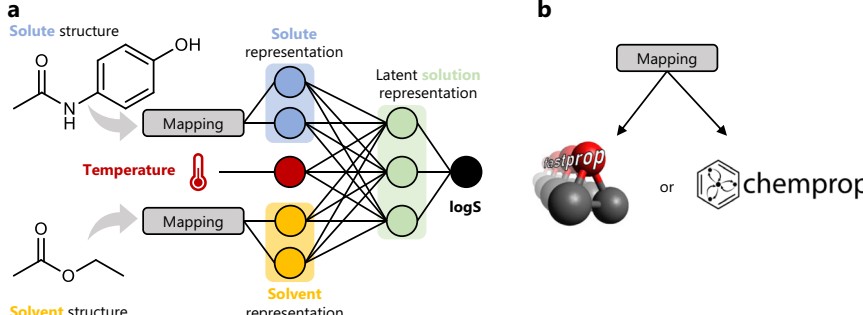

**Fig. 1 | Machine learning representation of solutions. a** In our modeling approach, solute (e.g. paracetamol) and solvent (e.g. ethyl acetate) structures are mapped to representation vectors. These representation vectors are concatenated to the solution temperature to arrive at a solution representation, which is passed into a fully-connected neural network and regressed to the log of solubility (S, mol L$^{-1}$). **b** Structures are mapped to feature vectors using a fixed representation of Mordred descriptors as implemented in FASTPROP, or a learned representation derived from message passing on a graph representation from CHEMPROP. We compare the performance of models trained on these fundamentally different solution representations.

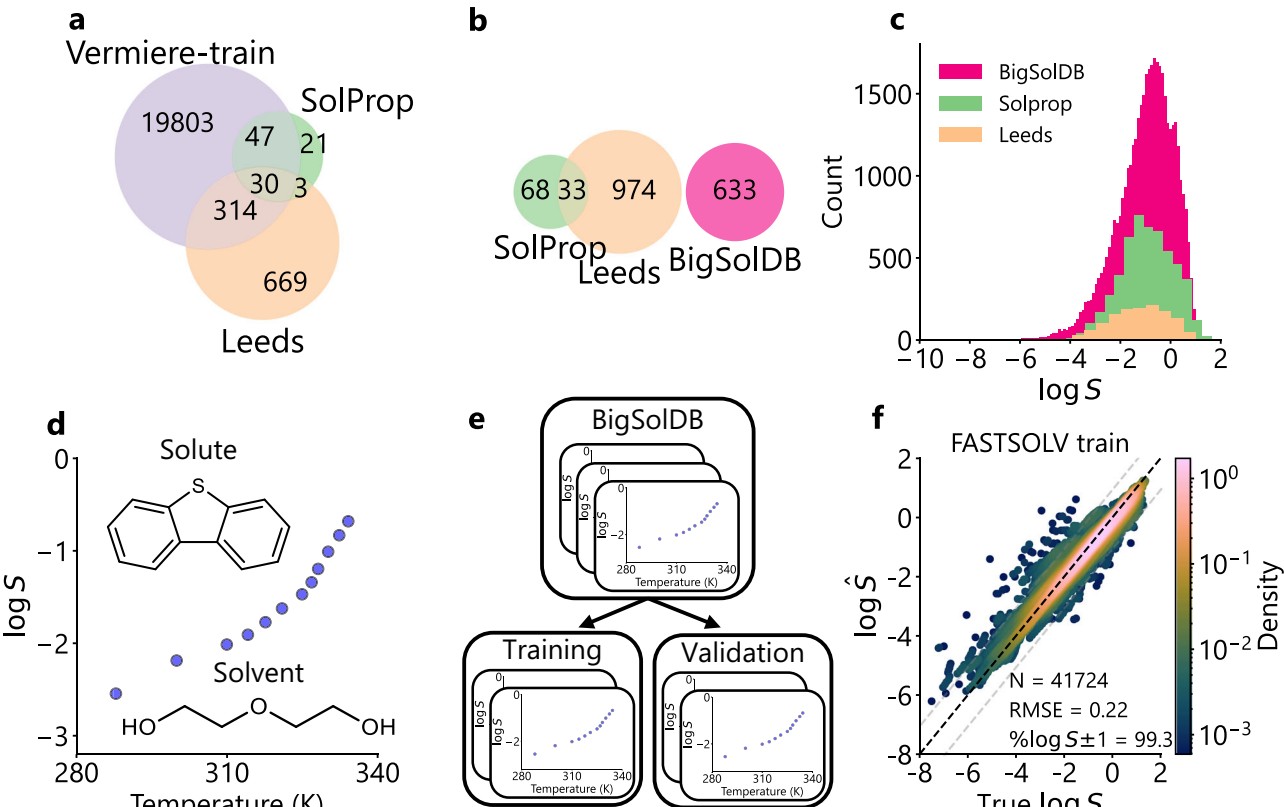

**Fig. 2 | Datasets and model training approach. a** The literature best Vermeire et al.[31] trained on a thermodynamic cycle with > 20k solutes. The solutes seen during training have overlaps with the SolProp and Leeds testing datasets. **b** In this study, we rigorously test solute extrapolation performance of our models by dropping overlap between the training dataset of BigSolDB[32], and the SolProp[31] and Leeds[24] testing sets. **c** Distribution of the label, the log of solubility (S, mol L⁻¹), across the training and external testing sets. **d** Demonstration of a single solubility experiment, which contains the measured solubility of a solute (dibenzothiophene) in a solvent (diethylene glycol) across a range of temperatures (K), as measured by

Tao et al.[58]. **e** Rigorous data splitting strategy splits training and validation data by experiment, ensuring data is not leaked during model selection. **f** Parity plot demonstrating predictions of optimized FASTPROP-based model on BigSolDB, including training and validation data (N = 41,724). The model achieves a root mean square error (RMSE) = 0.22 and % log S ± 1 = 99.3%. Black dotted line indicates parity line. Gray dotted lines indicate the bounds of log S ± 1. Color indicates density, determined using Gaussian kernel density estimation (KDE) visualized on a logarithmic scale. See Sections 3.1 and 3.2 for training details. Source data are provided as a Source Data file.

(Fig. 2b). This reflects our intended application—extrapolation into new solutes with no additional information. The Vermeire et al.[31] model has many overlapping solute structures in its training data (see Fig. 2a), leading to overly optimistic reported performance. To demonstrate this and provide a second point of comparison in which both the model developed by Vermeire et al.[31] and our models extrapolate to new solutes, we also tested on the Leeds organic solubility dataset prepared by Boobier et al.[24] This dataset is more diverse in solute structures than the SolProp dataset but contains solubility data only near room temperature. This makes it a rigorous test of a models' capacity to extrapolate to new chemical space, likely also with a higher aleatoric limit given the absence of de-facto averaging over multiple measurements as in the SolProp dataset. The Leeds dataset has less overlap in solute structures with the Vermeire et al.[31] training data than SolProp (Fig. 2a), making it a more stringent extrapolation test. Thus, testing the model developed by Vermeire et al.[31] and our models on the Leeds dataset evaluates their ability to handle diverse solute chemistry without considering the effects of temperature on solubility. The distribution of the label log S across all three datasets is similar, centered around -1 with a long tail in the limit of low solubility (Fig. 2c).

We trained our models using 95% of the remaining data in Big-SolDB, reserving 5% for validation and model selection. To avoid data leaks we split our data to ensure no solutes appear in both training and validation sets, meaning that every group of measurements for a

solute, one of which is visualized in Fig. 2d, will be grouped together. Since we split our dataset by experiment and solute (Fig. 2e), we ensure that we rigorously test extrapolation to new solutes. Additional details about model training and hyperparameter optimization can be found in Supporting Information Section S1. The performance of the FASTPROP-based model on training and validation is shown in the parity plot in Fig. 2f. Performance is quantified via the Root Mean Squared Error (RMSE), defined in Equation (1) as:

$$\text{RMSE} = \sqrt{\frac{1}{n}\sum_{i=1}^{n}(y_i - \hat{y}_i)^2}, \tag{1}$$

where $y_i$ is the true value, $\hat{y}_i$ is the predicted value, and $n$ is the number of observations. Additionally, the Percentage of Predictions within 1 log S unit, a metric based on the upper reported limit of experimental reproducibility, is referred to as % log S ± 1[24]. This is calculated in Equation (2) as:

$$\% \log S \pm 1 = 100 \times \frac{1}{n}\sum_{i=1}^{n}\mathbf{1}_{[|y_i-\hat{y}_i|\leq 1.0]} \tag{2}$$

where $y_i$ is the true value, $\hat{y}_i$ is the predicted value, $n$ is the number of observations, $\mathbf{1}_{[\cdot]}$ is the indicator function which equals 1 if the

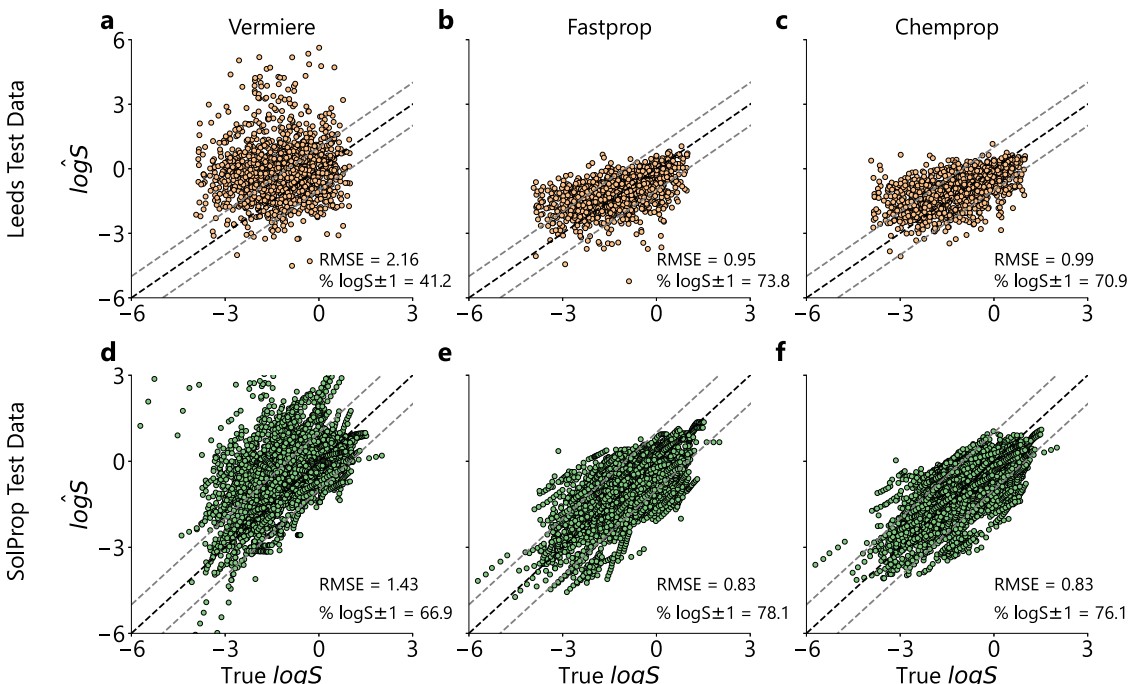

**Fig. 3 | Performance of literature best Vermiere model and our optimized solution FASTPROP and solution CHEMPROP models on test sets.** Parity plots of log of solubility (S, mol L$^{-1}$) on the x-axis against predicted log $\hat{S}$. Black dotted line indicates parity line. Gray dotted lines indicate the bounds of log $S \pm 1$ around the parity line. **a** Parity plot of the Vermiere et al.[31] model on the Leeds[24] test data, which has a root mean squared error (RMSE) = 2.16 and % log $S \pm 1$ = 41.2%. **b** Parity plots of the solution FASTPROP model on the Leeds[24] test data (RMSE = 0.95, % log $S \pm 1$ = 73.8%) and (**c**) the solution CHEMPROP on the Leeds[24] test data (RMSE = 0.99, % log $S \pm 1$ = 70.9%). Parity plots highlighting model predictions of (**d**) Vermeire et al.[31] (RMSE = 1.43, % log $S \pm 1$ = 66.9%), (**e**) solution FASTPROP (RMSE = 0.83, % log $S \pm 1$ = 78.1%), and (**f**) solution CHEMPROP (RMSE = 0.83, % log $S \pm 1$ = 76.1%) on the SolProp[31] dataset. Source data are provided as a Source Data file.

condition is true and 0 otherwise. The optimized solution FASTPROP model achieves excellent interpolation accuracy, with RMSE = 0.22 and % log $S \pm 1$ = 99.3%. Similarly, the optimized solution CHEMPROP model achieved an RMSE = 0.28 and % log $S \pm 1$ = 99.2%.

**Model performance on solute extrapolation**

After training solution FASTPROP and CHEMPROP models, their performance on extrapolation was evaluated on the Leeds and SolProp test sets and benchmarked against the Vermiere model as made available via a Python package in the original publication. We observe that the Vermiere model performs comparatively poorly on the Leeds dataset, with RMSE = 2.16 and % log $S \pm 1$ = 41.2% (Fig. 3a). On inspection there is observable systematic bias, with the model often over-predicting the solubility. In contrast, both the solution FASTPROP and CHEMPROP models perform similarly well, with RMSE = 0.95 and % log $S \pm 1$ = 73.8% for FASTPROP and RMSE = 0.99 and % log $S \pm 1$ = 70.9% for CHEMPROP (Fig. 3b, c). The systematic bias is greatly reduced in both models.

On the SolProp dataset, the Vermiere model performs slightly better (RMSE = 1.43 and % log $S \pm 1$ = 66.9%), but still exhibits systematic bias, with several specific experiments appearing with over-predicted temperatures gradients compared to parity (Fig. 3d). This performance is similar to what Vermiere et al.[31] reported when no experimental data was available and only molecular structures are used as inputs to the model. In contrast, the solution FASTPROP and CHEMPROP models perform significantly better with RMSE = 0.83 and % log $S \pm 1$ = 78.1% for FASTPROP and RMSE = 0.83 and % log $S \pm 1$ = 76.1% for CHEMPROP (Fig. 3e, f).

When running inference on both the Leeds and SolProp test sets, the solution FASTPROP model demonstrated inference times about fifty-fold faster than the Vermiere et al.[31] model due to its comparatively lightweight architecture. The use of molecular descriptors as an embedding also enabled analysis of results with SHAP[29] (Supporting Information Figure S1) to aid in interpretability of model results. We trained four FASTPROP models with different initializations and ensemble them, terming the resulting ensemble model FASTSOLV; further results in this study referencing FASTSOLV refer to this ensemble model specifically.

To go beyond these aggregate performance metrics and visually inspect prediction and gradient accuracy, we also evaluated our model performance on specific case studies from the SolProp test set (Fig. 4). We selected two structurally distinct solutes from the held-out SolProp test set–risperidone, a water-insoluble antipsychotic, and L-prolinamide, an amino acid amide–and compared model predictions from the FASTSOLV model and the Vermeire et al.[31] model against temperature-dependent experimental solubility data in different solvents. Risperidone was evaluated in polar acetone and isopropyl alcohol, while L-prolinamide was tested in highly nonpolar hexane and heptane.

We observe that FASTSOLV significantly improves absolute and gradient accuracy over the Vermeire et al.[31] model for risperidone and L-prolinamide solutions. Specifically, FASTSOLV achieves an RMSE of 0.16 for risperidone and 0.25 for L-prolinamide, compared to 1.64 and 2.33 for the Vermeire et al.[31] model. For risperidone, FASTSOLV correctly predicts the relative solubility order in acetone and isopropanol and predicts realistic temperature-dependence. The Vermeire et al.[31] model, in contrast, overpredicts solubility, has greater model uncertainty, and overestimates the gradient with respect to temperature (Fig. 4a). For L-prolinamide, FASTSOLV successfully discriminates solubility between the highly similar solvents hexane and heptane, correctly predicting higher solubility in heptane, while the Vermeire et al.[31] model predicts nearly identical values (Fig. 4b). The ability to rank solubility accurately and distinguish between structurally similar solvents is critical for high-throughput virtual screening and suggests that FASTOSLV accurately represents solvent structure.

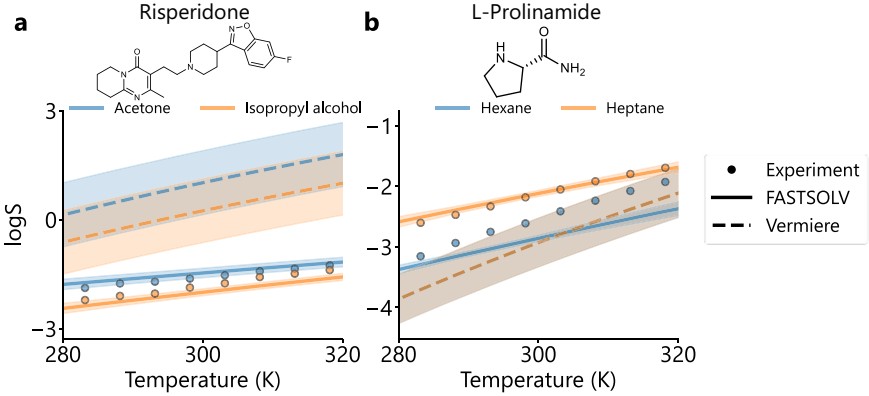

**Fig. 4 | Model validation on structurally different solutions.** Solubility predictions and experimental data as a function of temperature for solutions of (**a**) risperidone in acetone (blue) and isopropanol (orange) and (**b**) L-prolinamide in hexane (blue) and heptane (orange). Experimental data are plotted as circles, predictions from the Vermeire et al.[31] model are plotted as dotted lines, and ensemble average FASTSOLV predictions are plotted as solid lines. The shaded error bands indicate uncertainty in model predictions for both models. For FASTSOLV, uncertainty is the ensemble standard deviation. For the Vermeire et al.[31] model, uncertainty is propagated from uncertainty in predicted constitutive thermodynamic quantities. Experimental risperidone solubility data is compiled in SolPropVermeire et al.[31] from Mealey et al.[59]. Experimental L-prolinamide solubility data is compiled in SolProp from Cui et al.[60]. Source data are provided as a Source Data file.

We also sought to identify failure modes of FASTSOLV by probing solutes with inaccurate predictions. We observed poor prediction accuracy on anthraquinone solutions in acrylonitrile (RMSE = 1.80), methyl ethyl ketone (1.34), and isopropyl alcohol (1.20), using reference data from Cepeda and Diaz[38](Supporting Information Figure S2). Model performance on the parent molecule of anthraquinone, anthracene, in the same solvents exhibited much higher accuracy (RMSE = 0.42, 0.37, and 0.12 in acetonitrile, methyl ethyl ketone, and isopropyl alcohol, respectively). To further probe this poor performance, we also evaluated solubility on 85 solutions of the parent molecule anthracene and multiple -quinone derivatives in a variety of solvents which are compiled in the SolProp test set. We found an overall RMSE = 0.52, with RMSE = 0.76 for 4 anthracene solutions, 0.44 for 32 2-ethylanthraquinone solutions, and 0.55 for 49 1-chloroanthraquinone solutions (Supporting Information Section S3). However, even though the predictions on anthraquinone are relatively inaccurate, the model is still able to correctly rank order solubility between acetonitrile, methyl ethyl ketone, and isopropyl alcohol (Supporting Information Figure S2), suggesting the model has an accurate molecular representation of these challenging polycyclic aromatics.

## Model performance is capped by the aleatoric limit

As previously mentioned, there is substantial discussion in the literature about the inter-laboratory variability of solubility experiments, which can range between 0.5-1 log units, depending on the source and the variability metric used. We thus sought to establish if our model performance approaches this limit for the testing data. We first estimated the inter-laboratory experimental variability within the present datasets by identifying solutions with the same solute, solvent, and temperature but different literature sources; this yielded 34 solutions containing 8 unique solutes and 6 unique solvents. Between these solutions the average inter-laboratory standard deviation is 0.34 log while the RMSE is 0.75 log (Supporting Information Section S4). We use this value of inter-laboratory RMSE (0.75) for direct comparison with model performance, which also is evaluated in RMSE.

With this reference aleatoric limit established, we evaluated the model performance trajectory as a function of training dataset size. To do so, we randomly downsample the training dataset to some smaller size, train a four-model ensemble and report the performance on the test sets. Repeating this at different sizes of downsampled training sets with multiple replicates at each size generates a performance

trajectory as a function of training set size (Fig. 5a). We observe that the performance trajectories for both the FASTPROP and CHEMPROP models are similar, despite representing fundamentally different modeling approaches. We also observe that the model performance on the SolProp test set plateaus after only 500 experiments (-5000 data points) are included in training for both the CHEMPROP and FASTPROP models. Similarly, the performance on the Leeds test set plateaus for the CHEMPROP model after only 2000 experiments (-20,000 data points), although performance of the simpler FASTPROP model takes slightly longer to plateau.

Next, we compared model predictions on these solutions with multiple sources. Specifically, measurements of N-acetylglycine solubility in acetonitrile and methanol were compiled in BigSolDB from Zhao et al.[39] and SolProp from Guo et al.[40]. In acetonitrile, the experimental data differ drastically, and the model predicts solubility values in good agreement with the measurements from Zhao et al.[39] (Supporting Information Figure S3a). In methanol, the data from both sources are very similar, and the model predictions are in good agreement (Supporting Information Figure S3b).

It is possible that epistemic uncertainty from model inexpressiveness or error could cause this plateau in test performance. To investigate this point further we also benchmarked our models against state-of-the-art SMILES-based foundation models, MolFormer[41] and ChemBERTa-2[42] by fine-tuning on BigSolDB (Supporting Information Section S5). These models are dramatically different in fundamental architecture and number of parameters compared to FASTPROP and CHEMPROP. We observe that our models outperform both of these transformer models on the SolProp and Leeds test sets (Supporting Information Figure S4). This demonstrates that model testing performance is not limited by model inexpressiveness or low model capacity.

Interestingly, the FASTPROP and CHEMPROP predictions on the SolProp dataset are highly correlated, with a Pearson's $r = 0.81$ (Fig. 5b). This correlation is actually stronger than the correlation of either model's predictions of the SolProp testing set itself (0.66 for FASTPROP and 0.65 for CHEMPROP). Additionally, comparing the Cumulative Distribution Function (CDF) of predicted gradients of log $S$ with respect to temperature demonstrates the strong correlation between the FASTPROP and CHEMPROP model predictions (Fig. 5c). We observe that the Vermeire et al.[31] model has severe systematic error in $\frac{d \log S}{dT}$, achieving an Earth Mover's Distance (EMD) of 0.06. In contrast, our two models exhibit similar and highly accurate gradient

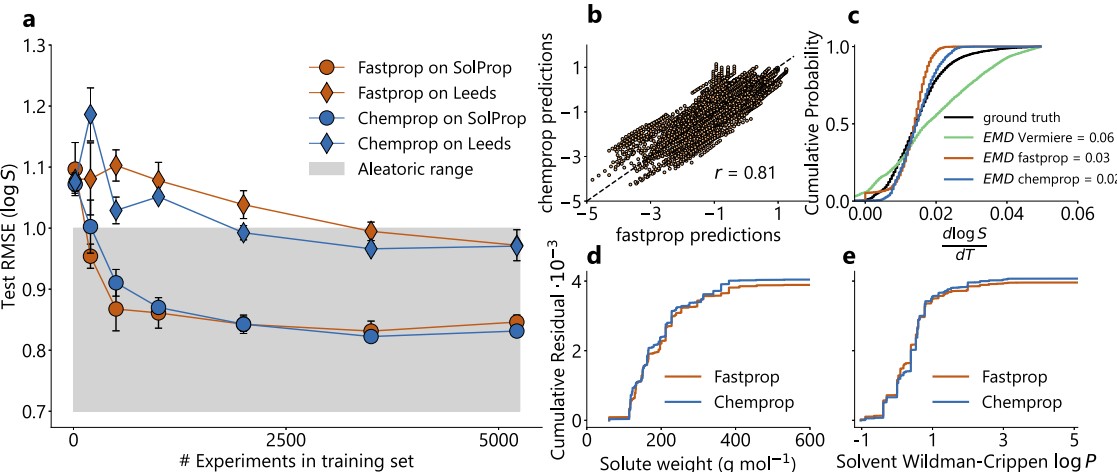

**Fig. 5 | Model performances reach the aleatoric limit. a** Mean test Root Mean Squared Error (RMSE) against the number of experiments used in the training dataset for solution FASTPROP and CHEMPROP on both the SolProp and Leeds datasets. Orange-colored curves show results for solution FASTPROP models, while blue-colored curves show test RMSE for solution CHEMPROP models. For both colors, diamond markers indicate test RMSE on the Leeds dataset, while circular markers indicate test RMSE on the SolProp dataset. The gray shaded area indicates the range of experimental uncertainties reported in literature ($0.5 < \log S < 1.0$). The plot cuts off at 0.7 for ease in visualizing the data. Error bars indicate standard deviation across three randomized training trials in which the subset of experiments used was randomly selected in order to account for variability in performance due to the effects of downsampling. **b** Correlation of FASTPROP and CHEMPROP model predictions on the SolProp[31] test set. One-sided Pearson's

correlation coefficient (r) was determined, with null hypothesis $r > 0$ and $N = 5700$. The resulting $r = 0.81$, with $p = 0.00$. The 95 % confidence interval of $r$ is [0.80, 1.00]. Black dotted line shows parity. **c** Cumulative Distribution Function (CDF) of the gradients of $\log S$ with respect to temperature $T$ ($\frac{d\log S}{dT}$) in the SolProp test set. Vermeire et al.[31] model predicted gradient CDF (green) achieves an earth movers distance (EMD) of 0.06 compared to the SolProp ground truth gradient CDF (black). FASTPROP predicted gradient CDF (orange, EMD = 0.03) and CHEMPROP predicted gradient CDF (blue, EMD = 0.02) compared to the SolProp ground truth gradient CDF (black). **d, e** Cumulative residual of FASTPROP (orange) and CHEMPROP (blue) model predictions of $\log S$ in the SolProp test set against (**d**) solute molecular weight (g/mol) and (**e**) solvent Wildman-Crippen $\log P$. Cumulative residual is multiplied by $10^{-3}$ for concise axis labels. Source data are provided as a Source Data file.

distributions, with EMD of 0.03 and 0.02. See Supporting Information Section S1 for further details on model training and gradient-based regularization. To further examine whether the models are predicting similar results, we plot the cumulative residuals of each model against important features in the SolProp test set. The cumulative residuals against solute molecular weight (Fig. 5d) and solvent Wildman-Crippen estimate of $\log P$[43] (Fig. 5e) both demonstrate that the FASTPROP and CHEMPROP models are making similar predictions and errors across the SolProp test set.

## Discussion

Here, we leveraged state-of-the-art cheminformatics software and large compiled solubility datasets to develop accurate and generalizable solubility models. Under strict extrapolation, we observe a 2-3 fold decrease in RMSE over the state-of-the-art model from Vermeire et al.[31]. In addition, the relative simplicity of the architecture decreases inference times up to fifty-fold. To the best of our knowledge, our models are the best performing solvent- and temperature-general models in the literature on solute extrapolation.

As shown in the solute Venn diagrams in Fig. 2, many entries in the SolProp testing set are present in the training set for the Vermeire et al.[31] model. This focus on interpolation during training leads to decreased performance in the extrapolation study shown in Fig. 3, where the performance decreased from an RMSE of 1.43 to 2.16 when moving from the SolProp to Leeds test set. Furthermore, we observe some nonphysical gradients with respect to temperature in the SolProp dataset, indicating that the model has not learned a comprehensive functional approximation of the temperature dependence of $\log S$.

In contrast, our optimized solution FASTPROP and solution CHEMPROP models exhibit much more consistent performance between the Leeds and SolProp test sets, highlighting the strong performance of our models under rigorous solute extrapolation (Fig. 3). The slightly decreased accuracy of our models on the Leeds test set is attributable to the increased solute diversity (Supporting Information

Figure S5) and lack of de-factor averaging from multiple temperature measurements, artifacts of the difference in its construction.

Both of our models also exhibit accurate predictions of the gradient on the solubility with respect to temperature (Figs. 4 and 5c, Supporting Information Figures S2 and S3), indicating that the models learned physically realistic temperature dependence, which is critically important in process chemistry applications[44,45]. This can be attributed to both the training procedure, which leverages gradient-based regularization (Supporting Information Section S1), and directly training on the BigSolDB dataset, which contains abundant temperature gradient data. Notably, unlike the Vermeire et al.[31] model, our models cannot take into account existing experimental reference data such as the free energy of solvation or Abraham solvation parameters, and in situations where such data is available, the Vermeire model could become more accurate.

Our evaluation of model performance on specific solutions also revealed that FASTSOLV accurately models temperature-dependence and correctly discriminates solubility between similar solvents, such as hexane and heptane (Fig. 4). Additionally, in cases where the model was accurate (Supporting Information Figure S2a–c), or relatively inaccurate (Supporting Information Figure S2d–f) in absolute performance, the model still correctly rank-ordered solubility in different solvents. Taken together, these results suggest that our model could readily be applied in high-throughput solvent screening workflows, which is crucial in synthetic[46] and crystallization[47] process development.

The training data size study and the error analysis presented in Fig. 5 show that our optimized models, which rely on distinct molecular mappings, both converge to the same performance limits with similar distributions of predictions and errors. The highly correlated predictions and cumulative residuals show both models achieve not only the same average performance, but predict similarly across the tests sets. Since CHEMPROP learns a molecular representation, this model should be able to continuously improve performance as the size of the training set increases, as demonstrated by both Heid et al.[48] for

several molecular properties and by Vermeire and Green[49] for solvation free energy. However, the test performance of our optimized CHEMPROP-based model stops improving after a relatively small amount of training data, at which point variability in the testing data prevents us from discerning improved model performance. As detailed in Section 1.3, we evaluated a measure of the experimental variability present between the training set and test sets, arriving at a standard deviation of 0.34, and an RMSE of 0.75 (Supporting Information Section S3). This standard deviation is well below the average standard deviation range reported in literature (0.5–0.7), suggesting this measure of experimental variability is likely a conservative estimate of the true aleatoric limit. However, our model performance on the test set approaches even this conservative estimate of the aleatoric limit, particularly for the SolProp test set (RMSE = 0.83 for both models). The comparatively poor performance of transformer-based foundation models MolFormer and ChemBERTa also support our conclusion that model expressivity is not limiting performance, but rather aleatoric uncertainty.

In the context of aqueous solubility, Palmer and Mitchell[16] compiled a curated set of highly accurate experimental measurements, concluding that model performance was limited by QSPR methods, rather than experimental variability. Since then, innovations in cheminformatics software have led to highly accurate and expressive model architectures which have been shown to perform excellently on molecular property prediction tasks[33,34]. We expect these flexible models to continue to improve as more training data are provided. However, as shown in Fig. 5a, there is no statistically significant reduction in test RMSE as thousands of additional data are added. These results instead suggest that experimental variability in the test data limits model performance and that better test data is needed to accurately assess the quality of our models. This affirms the observations in the Vermeire et al.[31] study, wherein the model was found to perform better on more accurate testing data. Current literature trends toward aggregating larger databases of published experimental solubility values. However, we have demonstrated that fewer than 500 experiments from our training set are needed to achieve near-optimal performance. Compiling larger training databases will not reduce test RMSE beyond the aleatoric limit demonstrated here. Future work should create accurate testing datasets of solubility in organic solvents in the same manner as the CheqSol dataset of highly accurate aqueous solubility compiled by Palmer and Mitchell[16] or apply careful, scientifically-informed curation as LLompart et al.[50] performed on AqSolDB[51]. Further innovations in model architecture or compiling more training data may improve predictions but they will be difficult to discern without better test data.

In conclusion, we present organic solubility prediction models using deep learning on fixed and learned molecular representations, then test them under rigorous solute extrapolation. Our models outperform comparable literature models by a factor of 2-3 on publicly-available test sets. We demonstrate that our models predict near the aleatoric limit of the experimental test data, motivating the assembly of highly accurate testing datasets for the field to notice further improvements in organic solubility prediction. Given the importance of solid solubility prediction, we have termed the fixed representation model FASTSOLV due to its rapid inference time and taken extensive steps to make it available. FASTSOLV can be accessed via any web browser for free at fastsolv.mit.edu, downloaded as a python package for use in scripting (pypi.org/project/fastsolv/), and is integrated directly within the ASKCOS (askcos.mit.edu) platform for retrosynthesis.

## Methods
### Training procedure
Rigorous evaluation of extrapolation requires careful preparation of the data for training, validation, and testing. Overlapping solute

structures in the testing and training data are dropped from the training data to avoid data leaks. The ASTARTES software package[52] is then used to randomly partition entire experiments from the training data into validation and training sets, again ensuring that no solute structures are seen by the model in both sets. A single model contains four individual networks trained on a different random training set to account for the affect of random sampling. Reported predictions on a given test set are thus the average prediction across these four trained models, and reported prediction uncertainty is the standard deviation of these predictions.

### Network architecture
Prior to arriving at the architecture shown in Fig. 1, more complex models inspired by the work of Pathak et al. were tested[53]. Extensive efforts were made to identify a physics-informed neural network architecture which would infuse inductive bias into the network and thereby improve predictions. Each of the following features was made available during the automated hyperparameter optimization such that the algorithm would automatically deduce which was the most effective:

- To allow the network to learn a unique per-solvent and per-solute representation rather than a single 'solution' representation, distinct linear layers could be added after the initial mapping for both the solute and the solvent. This reflects the intuitive understanding that each solute and solvent should have a unique contribution to the resulting solubility which is independent of the exact solution.
- The manner in which the latent solute and solvent representations are combined was also configurable, with some choices including elementwise addition, subtraction, multiplication, or simple concatenation. These are analogous to existing solubility prediction models such as the multiplicative Abraham model.

To explore the resulting massive design space, we leveraged the Optuna hyperparameter optimization framework of Akiba et al.[54] Across many repeated instances of optimization, the search algorithm always selected the comparatively simple architectures described in the present study. The addition of inductive bias was unable to surpass reliance on the comparatively simple architecture, at least given the current aleatoric limit of available data. See Supporting Information S1 for further details, including a complete table of the search space explored in this work.

Sobolev training[55] was implemented for both the Chemprop- and FASTPROP- based FASTSOLV models. This approach penalizes the network during training for both the error in the prediction of the solubility and the gradient of the predicted solubility with respect to the input temperature. The latter is approximated from the input data using finite differences, a reasonable approximation for the typically monotonic and locally-linear solubility curves. During training the gradient is found by continuing backpropagation through all network layers, as is usually done, and additionally the input temperature. The effect of Sobolev training is that networks generally converge in fewer epochs, have substantially increased accuracy relative to experimental gradients, and are stronger interpolators. The latter is of specific interest in some process applications of FASTSOLV. During inference, the FASTSOLV model will not continually increase predicted solubility past an input temperature of approximately 350 K. This is a deliberate design choice, given that many common organic solvents boil at or near this temperature: ethanol (351.5 K), benzene (353.2 K), acetonitrile (355.1 K), methyl ethyl ketone (352.8 K), hexane (341.8 K), tetrahydrofuran (339.1 K), and ethyl acetate (350.2 K).

### Aleatoric error study
The performance trajectory shown in Fig. 5 is generated by gradually increasing the amount of training data available to the model. The

model is trained in the same manner as described in Section 1.1, actually containing four separate networks trained on different random selections of the downsampled data. At first the model sees only a small number of experiments during training before subsequent testing on the holdout sets. The amount is gradually increased to the full size of the dataset, analogous to performing more solubility experiments to gather more samples in hopes of improving model performance.

## Reporting summary
Further information on research design is available in the Nature Portfolio Reporting Summary linked to this article.

## Data availability
We neither present nor generate original data as part of this study. Datasets used for training and testing within this study were retrieved from publicly available sources. The Boobier et al.[24] data used in this study are available in the Zenodo database under accession code 10.5281/zenodo.3686213. The Krasnov et al.[32] data used in this study are available in the Zenodo database under accession code 10.5281/zenodo.6984601. The Vermeire et al.[31] data used in this study are available in the Zenodo database under accession code 10.1021/jacs.2c01768. Further step-by-step instructions to retrieve the data and prepare it for training and testing are provided alongside the source code referenced in Code availability. Source data are provided with this paper.

## Code availability
The source code for model training, testing, and analysis is available on GitHub (https://github.com/JacksonBurns/fastsolv). A static snapshot of this code as used in this study has also been provided on Zenodo[56]. FASTSOLV is also packaged through PyPI and installable in Python via pip (https://pypi.org/project/fastsolv/). Model checkpoints are deposited on Zenodo[57]. FASTSOLV is also directly accesisble via a web interface (http://fastsolv.mit.edu/).

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

## Acknowledgements

This material is based upon work supported by the U.S. Department of Energy, Office of Science, Office of Advanced Scientific Computing Research, Department of Energy Computational Science Graduate Fellowship under Award Numbers DE-SC0023112 (J. B.) and DE-SC0022158 (L. A.). The authors thank Connor W. Coley for his helpful comments and suggestions during the formulation of this concept. The authors thank Florence Vermiere for helpful feedback and suggestions during the writing of this manuscript. The authors acknowledge the MIT SuperCloud and Lincoln Laboratory Supercomputing Center for providing HPC resources that have contributed to the research results reported within this article.

## Author contributions

LA: conceptualization (equal); methodology (equal); formal analysis (equal); software (supporting); writing - original draft (lead); review and editing (equal). JWB: conceptualization (equal); methodology (equal); formal analysis (equal); software (lead); writing - original draft (supporting); review and editing (equal). PSD: supervision (supporting); formal analysis (supporting); review and editing (supporting). WHG: supervision (lead); formal analysis (supporting); review and editing (supporting).

## Competing interests

The authors declare no competing interests.

## Additional information

**Peer review information** : *Nature Communications* thanks Seonah Kim and Farshud Sorourifar for their contribution to the peer review of this work. A peer review file is available.

