## [Transparent Peer Review file · Nature Communications]

Data-Driven Organic Solubility Prediction at the Limit of Aleatoric Uncertainty

Corresponding Author: Professor William Green

Version 0:

Reviewer comments:

Reviewer #1

(Remarks to the Author)

The manuscript addresses a significant challenge in chemical property modeling: the prediction of temperature dependent solubility in organic solvents. By leveraging modern machine learning architectures (fastprop and chemprop) and a large dataset (BigSolDB), the authors demonstrate promising improvements over existing models. The incorporation of extrapolation tests and open accessibility of the models via a Python package and web interface are commendable. The manuscript is well written, however, there are a few points that should be addressed before publication.

1. Aleatoric Limit Determination

- The explanation of how the aleatoric limit is determined could benefit from greater clarity. While references [12-17], [28], and [29] are cited, it is unclear if these works explicitly support the specific claim of a 0.5–1 logS uncertainty range. Direct evidence or a more detailed synthesis from these sources would strengthen the argument.
- Aleatoric uncertainty can also be heteroskedastic. Do the authors know if this is the case for organic solubility? Has a heteroskedastic Gaussian likelihood loss function been considered?
- If noise is heteroskedastic, the "aleatoric limit" of 0.5–1 logS may not apply uniformly across the dataset. This would mean the plateau in RMSE could be dataset-specific and not necessarily indicative of reaching a universal limit. .

2. Solute Diversity and Aleatoric Limit

- The statement, "The slightly decreased accuracy of our models on the Leeds test set is perhaps attributable to the increased solute diversity in the Leeds dataset," raises questions. If increased solute diversity reduces accuracy, then it suggests that the model predictions are not at the aleatoric limit. Could the authors clarify this apparent contradiction?

3. Use of Existing Experimental Data

An incomplete sentence states, "Notable though is that our models cannot take into account existing experimental reference data, and in situations where such data..." Please complete this thought.

Additionally, what specific features of the Vermeire model allow it to incorporate existing experimental reference data more effectively? Would fine-tuning the fastprop or chemprop models not yield similar benefits? A comparison of their flexibility in integrating external data would be insightful.

4. Alternative Explanations for RMSE Plateau

The manuscript posits that the plateau in test RMSE is due to reaching the aleatoric limit. However, alternative explanations, such as insufficiently descriptive molecular features or overly simple model architectures, are plausible. Could the authors provide evidence or reasoning to rule out these alternatives?

If the noise is heteroskedastic, do the authors still believe that, "While it is possible that further innovations in model architecture or compiling more training data can improve predictions, the improvements in model accuracy will not be discernible without better test data?"

5. Universal Approximation Theorem

The authors reference the universal approximation theorem (UAT) multiple times. Could they elaborate on their interpretation? The theorem is theoretical and doesn't directly address finite-capacity constraints or the ability of a model to generalize from finite data. Moreover, UAT doesn't guarantee good performance unless the model is trained with sufficient data that spans the entire domain of interest. The out of sample data used for extrapolation could limit the model's actual capacity, irrespective of theoretical considerations. How is the UAT applied practically in the context of fastprop and chemprop?

Grammar and Minor Issues:

- Broken citation on page 9—please correct or clarify.
- Consider proofreading for minor errors.

(Remarks on code availability)

Both the web interface and local version of fastsolv are functional and easy to use.

Using an in-house dataset consisting of experimental solubility measurements for 31 anthraquinone derivatives, I tested the model's extrapolation capabilities. The initial predictions exhibited a significant bias (RMSE = 3.96), but applying a simple linear correction reduced the RMSE to 0.50. This suggests that the model might struggle with certain solute classes without additional calibration.

While I recognize that the authors cannot verify or act on these specific results without access to my dataset, this observation highlights a potential area for future investigation.

I did not test other aspects of the code, aside from functionality and prediction accuracy. However, the repository is modular and sufficiently documented such that the work could be reproduced.

Reviewer #2

(Remarks to the Author)

The authors present a machine learning-based approach to solubility prediction. The authors use the precompiled BigSolDB dataset, consisting of variable temperature and variable solvent measurements of logS (thus incorporating the entirety of the thermodynamic cycle comprising the solvation process), to train and validate their model; model testing utilizes the Leeds solubility dataset compiled by Boobier et al. The authors compare the performance of a graph neural network-based (ChemProp) and an artificial neural network-based (FastProp) architecture, ultimately settling on the ANN-based architecture due to its relatively low computational overhead. The FastProp-based FastSolv architecture combines Mordred-generated descriptors for both solute and solvent, as well as a temperature embedding for model inputs, and consistently achieves an RMSE below 1 log unit on unseen test data. The authors note that this performance on the unseen test set is within the bounds of experimental uncertainty and posit that assessment of the model performance is thus hampered by the uncertainty endemic in the test set itself.

While the model's performance is admirable, it is the opinion of the reviewer that several points must be addressed prior to publication:

1. Incomplete introduction: The authors note several state-of-the-art neural network approaches for organic solubility prediction. Notably absent is the work by Kim et al., "Designing solvent systems using self-evolving solubility databases and graph neural networks", recently published in Chemical Science. This model achieves SOTA performance in predicting solvation free energies and generalizes well to unseen solvent-solute combinations. While the authors cite this work later in the text, they do so only to justify their choice of Vermiere et al.'s model as a metric of comparison. Neglecting to mention the

contribution of this work must be addressed.

2. Impact of Network Architecture on Model Performance: The authors provide extensive discussion in the text regarding model performance as the number of training data points is varied. This, in fact, is used by the authors to justify their choice of a relatively simple neural network architecture (“However, as we demonstrated, only a small subset, 5000 points, of our training set are needed to achieve near-optimal performance” page 9; “Simply relying on Universal Approximation theorem is sufficient – additional inductive bias does not improve performance”, page 10). However, the authors do not support this claim with meaningful experiments. In particular, the trend shown in Figure 4 suggests that information bottlenecks may be present for both the GNN-based ChemProp and the ANN-based FastProp architectures. Analysis of the per-layer latent space representations would clarify this and should be included given the significance of this “Sutton’s bitter pill” argument on experimental methodology.

3. Descriptor Selection in FastProp Architecture: FastProp utilizes the Mordred software to calculate molecular descriptors as inputs to the neural network architecture. However, the manuscript seemingly does not provide an analysis of the most relevant descriptors. Furthermore, several of the >1700 Mordred descriptors are highly correlated – no analysis of input descriptor correlation is performed. Without correlation and significance analysis, chemical insights from model training are impossible to draw, and model utility is significantly reduced

4. Clarification on Aleatoric Limit and Experimental Variability: Authors state that prior works indicate the range of experimental for solubility typically spans 0.5 - 1.0 log S. But it’s unsure how you derived the conclusion that both fastprop and chemprop predictions have essentially reached that “limit.” Did the authors specifically quantify the “noise floor” of the compiled dataset (e.g., by looking at repeated measurements for the same solute-solvent pairs)? More emphasis on the distinction between ‘error from data variability’ vs. ‘model capacity error’ would further strengthen the discussion around the irreducible error.

5. Data Quality vs. Data Quantity: Authors demonstrate that adding more data beyond certain data points (5000 / 20000) does not improve performance significantly. However, it hasn’t been discussed whether the quality of data in BigSolDB might dramatically vary across sources. For instance, do authors observe or suspect systematic biases from certain literature sources? Could cleaning or curating subsets of BigSolDB further improve predictions? This ties into the written conclusion that more carefully curated datasets (like the CheqSol approach for aqueous solubility) might be necessary for further performance gains.

6. The gradient of Solubility vs. Temperature: Authors show that Vermeire et. al.’s model systematically deviates in some instances. While the authors provide a cumulative distribution plot, the physical realism of the predicted gradients is never shown – especially in highly non-ideal solutions or near precipitation points. Do authors see any domain of temperature or solvent-solute classes (e.g., strongly hydrogen-bonding solvents or highly non-polar solvents) where the model might deviate from physically expected behavior?

It is thus the opinion of the reviewer that this work be published in Nature Comms pending major revisions

(Remarks on code availability)

The GitHub repository associated with the paper is well-structured, with clear organization of files and directories. It provides a detailed README file that includes comprehensive instructions for installing and running the model, making it accessible for users in the community to reproduce the results.

However, one issue encountered was with the web-based version of the fastsolv predictor, fastsolv_web, hosted at fastsolv.mit.edu. Attempting to access this resource consistently raised an internal server error. This may hinder users who wish to explore the tool online, especially those without the expertise or resources to run the application locally. Addressing this issue would greatly enhance the accessibility and usability of the tool for the broader community.

Overall, while the repository itself is a valuable resource, ensuring that the web-based version is functional would further support the reproducibility and practical application of the work.

Version 1:

Reviewer comments:

Reviewer #1

(Remarks to the Author)

The responses to the questions and revisions to the manuscript have addressed all my concerns, and is now suitable for publication.

(Remarks on code availability)

Yes, the code is available. It includes the model classes and links to the model weights, making the results reproducible; however, I did not attempt to reproduce the results. The repository also includes a user friendly code with examples that I tested, and a simplified web tool for broader community usage.

Reviewer #2

(Remarks to the Author)

The authors have satisfactorily addressed the points raised in the review. While the inclusion of a per-layer latent space analysis would add to the arguments made in the manuscript, it does present the addition of scope that may be beyond this manuscript (Point 2 in the original review). Thus, it is the opinion of the reviewer that the manuscript is suitable for publication in its revised form.

(Remarks on code availability)

We thank the reviewers for their thorough and insightful comments regarding our manuscript, as well as their recommendation that the manuscript be accepted for publication following revisions. Said revisions have been completed, with point-by-point responses to individual comments from the reviewers shown below.

Reviewer #1 (Remarks to the Author):

The manuscript addresses a significant challenge in chemical property modeling: the prediction of temperature dependent solubility in organic solvents. By leveraging modern machine learning architectures (fastprop and chemprop) and a large dataset (BigSolDB), the authors demonstrate promising improvements over existing models. The incorporation of extrapolation tests and open accessibility of the models via a Python package and web interface are commendable. The manuscript is well written, however, there are a few points that should be addressed before publication.

1. Aleatoric Limit Determination

- The explanation of how the aleatoric limit is determined could benefit from greater clarity. While references [12-17], [28], and [29] are cited, it is unclear if these works explicitly support the specific claim of a 0.5–1 logS uncertainty range. Direct evidence or a more detailed synthesis from these sources would strengthen the argument.

Response:

We agree that the discussion of the aleatoric limit in the original manuscript should be strengthened with more evidence and clarity. In the revised manuscript we have added substantial detail and evidence explaining how the aleatoric limit has been determined. First, we specifically extracted the aleatoric limit for aqueous solutions suggested by references. The cited sources report aleatoric limits between 0.5-1 log unit, depending on the measure of variability (standard deviation, RMSE, range). Most sources have converged on 0.5-0.7 log unit in standard deviation for aqueous solutions. However, solubility measurement in organic solvents is less standardized, involves a huge variety of solvents, and is complicated by solvent volatility. As a result there is less agreement on the aleatoric limit for solubility in organic solvents. As we emphasize in the introduction, there is not a universal aleatoric limit, but each compiled dataset has its own aleatoric limit based on the irreducible experimental variability in its constituent data. We have added the specific limits suggested by the cited references and added additional context to the **third paragraph in the introduction**.

Additionally, we have gathered direct evidence for the inter-laboratory experimental variability from our datasets. We looked at solutions with the same solute, solvent, and temperatures that overlapped between each pair of datasets (SolProp, BigSol, and Leeds), but were compiled from different sources. We found 34 overlapping solubilities points compiled from different sources, containing 8 unique solutes, and 6 unique solvents, with an overall inter-laboratory

experimental variability RMSE = 0.75 and standard deviation = 0.34. This direct evidence from our datasets is likely a conservative estimate, given the standard deviation is only 0.34, but gives us a reference point against which to benchmark our model RMSE. These results have been included in the **first and third paragraph of Section 1.3**. Correspondingly, we discussed how this finding of a measure of experimental variability in the data supports our conclusion that the model is approaching this performance limit in the **sixth paragraph of the discussion, Section 2**.

Within this subset, there were two solutions from different sources that had several temperature points. These were both solutions of N-acetylglycine, in acetonitrile and methanol, from references 41 and 42. We thus evaluated model performance against the different experimental results, with the caveat that we are not in the position to determine which experimental source is “correct.” In acetonitrile, the two experimental sources were different by about 1 log unit, while in methanol, the experimental sources were nearly overlapping. We included this analysis in **Supporting Information Section S4**.

- Aleatoric uncertainty can also be heteroskedastic. Do the authors know if this is the case for organic solubility? Has a heteroskedastic Gaussian likelihood loss function been considered?

Response:

We agree that it is likely that the aleatoric uncertainty is heteroskedastic with respect to solubility. Our hypothesis is that the experimental variability should be U-shaped, with higher uncertainty in the limits of low solubility and high solubility. In the limit of low solubility, it can be challenging to separate small undissolved aggregates from solution, leading to apparent solubilities higher than the true values, while long dissolution times can lead to underestimated solubility measurements.¹ In the limit of high solubility, differences between solute solid-state structures (polymorphs, salts, hydrates) can lead to magnified differences in measured solubility or even supersaturation. We thus expect that the uncertainty, in theory, should be U-shaped. In addition, surely different measurement methods employed in different labs have different errors.

Unfortunately we don't have enough data to comment about the heteroskedasticity of the experimental variability. For the sake of completeness, we plotted the distribution of the experimental variability in RMSE for the experiments in our small (N=34) lab-to-lab variability set (see below). We do observe some heteroskedasticity in the distribution of experimental variability which matches our intuition- the variability is higher for lower solubility. We don't have any data in the regime of higher solubility. Again for such a small sample size, we are reluctant to offer a further comment on this in our manuscript. Also, analyses of the experimental errors are best done by researchers with in-depth knowledge of the experimental protocols and problems encountered during experiments. We do not have that expertise.

We appreciate the recommendation of considering a heteroskedastic gaussian loss, which we have not considered. While this is an interesting idea, we believe that it is out of scope of the current manuscript, but could be a fruitful area of follow-up.

- If noise is heteroskedastic, the "aleatoric limit" of 0.5–1 logS may not apply uniformly across the dataset. This would mean the plateau in RMSE could be dataset-specific and not necessarily indicative of reaching a universal limit.

Following the discussion above, it is possible and potentially likely that the aleatoric limit does not apply uniformly across the dataset, since the experimental variability is potentially heteroskedastic. We plotted the distribution of the model prediction error on the SolProp dataset over the solubility range below, for both the fastprop- and chemprop-based models. Indeed, we do see that the error distribution follows a slightly U-shaped curve, which follows the heteroskedasticity we expect for the experimental variability. Again, since we don't have enough data to confirm the heteroskedasticity of the experimental variability, we are unable to confidently confirm whether the model error distribution is U-shaped because of the U-shaped aleatoric limit, rather than due to the problems in the model near the high or low solubility limits (i.e. epistemic error)

Additionally, we do believe that regardless of the true shape of the experimental variability distribution, each dataset will have its own aleatoric limit. This is because the exact sources of experimental variability are not necessarily consistent between datasets. It is not merely gaussian measurement noise that leads to this inter-laboratory variability, but bias in measurement due to poor control over solid state structure, inconsistent temperature control, and different methods of data analysis.² Thus, each compiled dataset may have a different aleatoric limit, and there is not necessarily a universal limit. We have clarified this point in the **third paragraph of Section 1.1**.

2. Solute Diversity and Aleatoric Limit

- The statement, "The slightly decreased accuracy of our models on the Leeds test set is perhaps attributable to the increased solute diversity in the Leeds dataset," raises questions. If increased solute diversity reduces accuracy, then it suggests that the model predictions are not at the aleatoric limit. Could the authors clarify this apparent contradiction?

Response:

We have revised **Section 1.1** and **Section 2** to emphasize that the "aleatoric limit" quantity which we reference in the manuscript is meant to be interpreted on a per-dataset basis rather than on a universal per-quantity (i.e. solubility as a whole) basis. The **second paragraph of the introduction** has also been significantly revised to state both the experimental considerations that lead to error and the range of error values reported in the literature.

We also note that the construction of the Leeds dataset is a contributor to the difference in performance. The SolProp test dataset has a de-facto averaging effect because of the presence of multiple measurements for each solution (at slightly different temperature), whereas the single-temperature measurements of the Leeds dataset do not. Additionally, the single-measurement nature of the Leeds dataset causes increased solute diversity in general, again by construction. This increased solute diversity indicates that our models will be extrapolating in chemical space further from their training data, which is also possibly a contributor to worse performance. To demonstrate that the solutes in the Leeds dataset have broader coverage in chemical space, we plotted a 2-D projection using UMAP, which allows for easy visualization of the much broader solute coverage of Leeds compared to the solutes in the training set. This analysis is now included in **Supporting Information Section S6**.

3. Use of Existing Experimental Data

An incomplete sentence states, "Notable though is that our models cannot take into account existing experimental reference data, and in situations where such data..." Please complete this thought. Additionally, what specific features of the Vermeire model allow it to incorporate existing experimental reference data more effectively? Would fine-tuning the fastprop or chemprop models not yield similar benefits? A comparison of their flexibility in integrating external data would be insightful.

Response:

We thank the reviewer for noticing the error in the incomplete sentence; the sentence has been corrected.

Additionally, the manuscript has been revised to better explain why the Vermeire et al. model is able to leverage existing experimental data whereas fastsolv cannot. In short, the Vermeire et al. model is a collection of machine learning models which each predict the Gibbs free energy, the enthalpy of solvation, and the Abraham solvation parameters, then combines them via a thermodynamic cycle to predict the solubility. If any of these values are known to high accuracy they can be used in place of the machine learning-based predictions, improving performance. We have clarified this point in the **fourth paragraph of the introduction** and the **fourth paragraph of Section 2**.

On the other hand, fastsolv predicts solubility directly from SMILES. At no point does it include any quantities which could be replaced with known higher quality values. For this reason, fine-tuning is not possible without retraining. The corrected sentence in the **fourth paragraph of Section 2** acknowledges this advantage of the Vermeire approach. However, the simplicity of our model is an advantage for our target applications, like high-throughput screenings in discovery pipelines and process development, where no higher-quality values are readily available. Finally, our overall improved accuracy over the Vermeire approach supports the decision to not rely on these reference data.

4. Alternative Explanations for RMSE Plateau

The manuscript posits that the plateau in test RMSE is due to reaching the aleatoric limit. However, alternative explanations, such as insufficiently descriptive molecular features or overly simple model architectures, are plausible. Could the authors provide evidence or reasoning to rule out these alternatives? If the noise is heteroskedastic, do the authors still believe that, "While it is possible that further innovations in model architecture or compiling more training data can improve predictions, the improvements in model accuracy will not be discernible without better test data?"

Response:

We agree with the reviewer's intuition that an insufficiently expressive descriptor set or model architecture could cause a test RMSE plateau, but we have provided substantial evidence as to why we believe those are not the source of the RMSE plateau in our case. First, regarding the molecular features, we suggest that the performance of the fastprop (descriptor-based) model plateauing at the same point as the chemprop (learned representation-based) model indicates that the descriptor set is at least *as expressive* as the representation learned by the chemprop model. In theory, the chemprop model can learn features that are as accurate/expressive as possible, which could be more descriptive than the fixed descriptor set used in fastprop. This indicates that the RMSE plateau is likely not due to the insufficiently descriptive molecular features. Discussion of this concept is included in the **sixth paragraph of Section 2**.

Regarding the possibility of overly simple model architectures leading to this plateau, we further emphasize the extensive hyperparameter optimization we performed with regard only to performance on randomly selected validation sets. This allowed the optimized models to be as large and expressive as required to achieve optimal performance. Beyond model size and expressiveness, various forms of inductive bias were available to the hyperparameter optimization algorithm. The search space included additional latent network layers for the solute and solvent to learn separate representations before combining them into the solution representation. Additionally, the search space included a configurable interaction operation (i.e. multiplication, addition, or concatenation) to combine the solute and solvent representations. This is similar to approaches that others, like Pathak et al., have taken to include physical intuition into prediction of solvation-based properties.³ For example, multiplication of solvent and solute learned features is analogous to the intuition of an Abraham solvation model, while addition is analogous to a group-additivity model. However, the result of many hyperparameter optimization instances was consistently that the simplified architecture presented in the manuscript was the most performant. **Section S1** has been added to the **Supporting Information** to provide a detailed description of the hyperparameter optimization and the breadth of model architectures that were considered.

To provide further evidence that our accuracy is not limited by model simplicity, we fine-tuned two state-of-the-art transformer models, MolFormer⁴ and ChemBERTa-2,⁵ which are designed for molecular property prediction, on this solubility task. We fine-tune these models using the same training set used to train the fastprop- and chemprop-based models reported in our study,

then test on the SolProp and Leeds datasets. The code to reproduce the fine-tuning can be found in our GitHub repository, linked here. The results of this benchmarking study are summarized in the **fourth paragraph of Section 1.3** and detailed in **Supporting Information Section S5**, where we report that these models under-performed relative to the models we developed.

Finally, even if a hypothetical improved (i.e. larger and or more accurate) test set had heteroskedastic noise, we would still be able to detect improvement as long as noise was reduced across the whole domain. As demonstrated in the error distributions provided above in our response to question 1, our models have learned a heteroskedastic noise distribution, which likely (but indeterminately) would match the heteroskedastic variability of the experimental data.

The combination of these sources of evidence indicates that the performance has reached the aleatoric limit of the test data. Were the test data more accurate, it may be possible to detect a difference between our two models and attribute the plateau to one of the aforementioned shortcomings.

5. Universal Approximation Theorem

The authors reference the universal approximation theorem (UAT) multiple times. Could they elaborate on their interpretation? The theorem is theoretical and doesn't directly address finite-capacity constraints or the ability of a model to generalize from finite data. Moreover, UAT doesn't guarantee good performance unless the model is trained with sufficient data that spans the entire domain of interest. The out of sample data used for extrapolation could limit the model's actual capacity, irrespective of theoretical considerations. How is the UAT applied practically in the context of fastprop and chemprop?

Response:

We thank the reviewer for pointing out this deficiency in our analysis of the results. Our reliance on the purely theoretical UAT was removed and replaced in **Section 3.1.1** and in **Supporting Information Section S1**. We instead emphasize the extensive hyperparameter optimization we performed, provide more in-depth discussion around the per-dataset aleatoric limit, and rely on performance results across diverse model architectures beyond those originally studied (see **Reviewer 1, Question 4**). Critically, as discussed in **Section 3.1.1** and **Supporting Information Section S1**, we leveraged state-of-the-art optimization tools to explore a design space that encompassed large models with built-in inductive bias. Repeated optimizations revealed that the comparatively simple architecture presented in the manuscript was optimal, leading us to the aforementioned conclusion.

6. Grammar and Minor Issues:

- Broken citation on page 9—please correct or clarify.
- Consider proofreading for minor errors.

Response:

We appreciate these careful observations. The citation on prev. page 9 (now page 12) is fixed, and is numbered as reference 34. We have also fixed several minor grammatical and spelling errors, which can be detected in the revised manuscript.

7. Reviewer #1 (Remarks on code availability): Both the web interface and local version of fastsolv are functional and easy to use.

Using an in-house dataset consisting of experimental solubility measurements for 31 anthraquinone derivatives, I tested the model's extrapolation capabilities. The initial predictions exhibited a significant bias (RMSE = 3.96), but applying a simple linear correction reduced the RMSE to 0.50. This suggests that the model might struggle with certain solute classes without additional calibration.

While I recognize that the authors cannot verify or act on these specific results without access to my dataset, this observation highlights a potential area for future investigation.

I did not test other aspects of the code, aside from functionality and prediction accuracy. However, the repository is modular and sufficiently documented such that the work could be reproduced.

Response:

We appreciate the reviewer ensuring the reproducibility, modularity, and functionality of fastsolv.

We also appreciate testing fastsolv on in-house experimental data. The need for a constant offset to correct for the stated bias is discouraging, though the presence of significant systematic error in our training data (as mentioned in **the second paragraph of the introduction**) may be the driving cause. Unfortunately, as the reviewer mentioned, we cannot verify these results for anthraquinone derivatives directly.

However, we did find 85 examples of anthracene (the parent molecule of anthraquinone) and two anthraquinone derivatives, 2-ethylanthraquinone, and 1-chloroanthraquinone, in 53 unique solvents within the SolProp test set, all at room temperature. Testing fastsolv on this SolProp subset yields an overall RMSE = 0.52, with RMSE = 0.76 for 4 anthracene solutions, RMSE = 0.44 for 32 2-ethylanthraquinone solutions, and RMSE = 0.55 for 49 1-chloroanthraquinone solutions. These results suggest the model is generally accurate on these potentially challenging polycyclic aromatic compounds.

To extend this further, we found another published solubility dataset for anthraquinone and anthracene, in acetonitrile (ACN), methyl ethyl ketone (MEK), and isopropyl alcohol (IPA).⁵ Overall, we observe highly accurate predictions on the parent molecule anthracene, with RMSE = 0.42, 0.37, and 0.12 for ACN, MEK, and IPA respectively. However, the model is much less accurate on anthraquinone, with RMSE = 1.80, 1.34, and 1.20 for ACN, MEK, and IPA respectively. Matching the reviewer's analysis, the predictions seem to be offset by a constant value here as well (Supporting Information Figure S3). This suggests that the model may struggle

with anthraquinone, despite being accurate for other polycyclic aromatics (as demonstrated above). However, even for anthraquinone, we still observe highly accurate temperature gradients and correct solvent ordering (ACN>MEK>IPA), which suggests the model still learned an adequate enough representation of anthraquinone to capture relative solubilities. This means the model could still be useful for solvent screening tasks even for solutes where absolute predictions are less accurate. We agree it could be a useful area of future exploration to determine why exactly anthraquinone challenges the model's accuracy.

This analysis is included in **Supporting Information Section S3** and the **sixth paragraph in Section 1.2**.

Reviewer #2 (Remarks to the Author):

The authors present a machine learning-based approach to solubility prediction. The authors use the precompiled BigSolDB dataset, consisting of variable temperature and variable solvent measurements of logS (thus incorporating the entirety of the thermodynamic cycle comprising the solvation process), to train and validate their model; model testing utilizes the Leeds solubility dataset compiled by Boobier et al. The authors compare the performance of a graph neural network-based (ChemProp) and an artificial neural network-based (FastProp) architecture, ultimately settling on the ANN-based architecture due to its relatively low computational overhead. The FastProp-based FastSolv architecture combines Mordred-generated descriptors for both solute and solvent, as well as a temperature embedding for model inputs, and consistently achieves an RMSE below 1 log unit on unseen test data. The authors note that this performance on the unseen test set is within the bounds of experimental uncertainty and posit that assessment of the model performance is thus hampered by the uncertainty endemic in the test set itself.

While the model's performance is admirable, it is the opinion of the reviewer that several points must be addressed prior to publication:

1. Incomplete introduction: The authors note several state-of-the-art neural network approaches for organic solubility prediction. Notably absent is the work by Kim et al., "Designing solvent systems using self-evolving solubility databases and graph neural networks", recently published in Chemical Science. This model achieves SOTA performance in predicting solvation free energies and generalizes well to unseen solvent-solute combinations. While the authors cite this work later in the text, they do so only to justify their choice of Vermiere et al.'s model as a metric of comparison. Neglecting to mention the contribution of this work must be addressed.

Response:

We appreciate the suggestion to reference this improved solvation free energy predictor, and have revised **Section 1.1** to include a reference to this work. We decided not to use this model

to supplant the predictor included within the Vermeire et al. thermocycle framework for the sake of fair comparisons against the original Vermeire et al. model. This justification is included in the **second paragraph of Section 1.1**.

2. Impact of Network Architecture on Model Performance: The authors provide extensive discussion in the text regarding model performance as the number of training data points is varied. This, in fact, is used by the authors to justify their choice of a relatively simple neural network architecture (“However, as we demonstrated, only a small subset, 5000 points, of our training set are needed to achieve near-optimal performance” page 9; “Simply relying on Universal Approximation theorem is sufficient – additional inductive bias does not improve performance”, page 10). However, the authors do not support this claim with meaningful experiments. In particular, the trend shown in Figure 4 suggests that information bottlenecks may be present for both the GNN-based ChemProp and the ANN-based FastProp architectures. Analysis of the per-layer latent space representations would clarify this and should be included given the significance of this “Sutton’s bitter pill” argument on experimental methodology.

Response:

We agree with the reviewer’s assertion that the original manuscript did not present sufficient evidence to support the claims relating to model capacity limitations causing information bottlenecks. The manuscript has been extensively revised to emphasize the significant efforts which were made during the initial study to create a larger and more expressive model, especially in **Section 3.1** and **Supporting Information S1**. In short, hyperparameter optimization revealed that the architecture presented in the manuscript was optimal in comparison to larger models incorporating inductive bias. Additional benchmarking with state-of-the-art transformer models was also added, and further demonstrated evidence of our conclusion. Our response to **Reviewer 1, Question 4** may also be informative in regard to this suggestion.

We have not performed a per-layer latent space analysis of the models trained for this study. While this is a useful suggestion, we believe it would represent a conceptual advance that is out of scope for the present work. We believe that the extensive exploration of design space described above indicates that the chosen model architecture is not suffering from an information bottleneck. We also suggest that the results of SHAP analysis (see **Question 3** below) offer further evidence that the representation learned by fastprop is sufficient.

3. Descriptor Selection in FastProp Architecture: FastProp utilizes the Mordred software to calculate molecular descriptors as inputs to the neural network architecture. However, the manuscript seemingly does not provide an analysis of the most relevant descriptors. Furthermore, several of the >1700 Mordred descriptors are highly correlated – no analysis of input descriptor correlation is performed. Without correlation and significance analysis, chemical insights from model training are impossible to draw, and model utility is significantly reduced

Response:

We thank the reviewer for pointing out this substantial omission. We have added **Section S2 to the Supporting Information** to demonstrate SHAP value analysis of the trained fastsolv model on the SolProp and Leeds test sets. In short, the SHAP plots show that despite molecular descriptors being highly correlated, the representation used in fastsolv is highly similar across different test datasets and leverages ~70% of the available descriptors, indicating it is not simply a degenerate or spurious representation. They also demonstrate the significance of a subset of molecular descriptors for the solute, which could help guide solute design and gain chemical insight, and reinforce our intuitive understanding that solubility is a strong function of temperature

4. Clarification on Aleatoric Limit and Experimental Variability: Authors state that prior works indicate the range of experimental for solubility typically spans 0.5 - 1.0 log S. But it's unsure how you derived the conclusion that both fastprop and chemprop predictions have essentially reached that "limit." Did the authors specifically quantify the "noise floor" of the compiled dataset (e.g., by looking at repeated measurements for the same solute-solvent pairs)? More emphasis on the distinction between 'error from data variability' vs. 'model capacity error' would further strengthen the discussion around the irreducible error.

Response:

We agree. Reviewer 1 also raised a very similar point. Please see our response to **Reviewer 1, Questions 1, 2, and 4**, and the corresponding revisions to the manuscript.

5. Data Quality vs. Data Quantity: Authors demonstrate that adding more data beyond certain data points (5000 / 20000) does not improve performance significantly. However, it hasn't been discussed whether the quality of data in BigSolDB might dramatically vary across sources. For instance, do authors observe or suspect systematic biases from certain literature sources? Could cleaning or curating subsets of BigSolDB further improve predictions? This ties into the written conclusion that more carefully curated datasets (like the CheqSol approach for aqueous solubility) might be necessary for further performance gains.

Response:

We agree that data "quality", both in terms of coverage of chemical space and the experimental accuracy of the data selected could influence performance when training on a smaller, down-selected training set. This issue is partially mitigated since we perform three replicates where we randomly select three different subsets and train different models on each. The data points plotted in **Figure 5a** (prev. Figure 4a) represents the average RMSE values across these three replicates, and the error bar represents the standard deviation. These different subsets represent different sampling of BigSolDB, and would have different experimental accuracy and different coverage over chemical space. We do observe some modest variation in model performance across models trained with these different subsets, though the effect shrinks with

increasing subset size as these effects average out. The **second paragraph of Section 1.3** has been modified to reflect that this downsampling experiment reflects random sampling.

While cleaning or curating subsets of BigSolDB could potentially reduce the effect of experimental variability in the training data, hand curation of BigSolDB would require substantial effort, and is out of scope for this work. Entire published works have been devoted to hand curation and the effect of such curation on model performance, such as work from Meng et al. and Llompart et al. on aqueous solubility databases.^{7,8} We have modified the call to action in the **seventh paragraph of Section 2** to more clearly encourage careful curation of organic solubility databases that could unlock the ability to evaluate improved model predictions:

6. The gradient of Solubility vs. Temperature: Authors show that Vermeire et. al.'s model systematically deviates in some instances. While the authors provide a cumulative distribution plot, the physical realism of the predicted gradients is never shown – especially in highly non-ideal solutions or near precipitation points. Do authors see any domain of temperature or solvent-solute classes (e.g., strongly hydrogen-bonding solvents or highly non-polar solvents) where the model might deviate from physically expected behavior?

It is thus the opinion of the reviewer that this work be published in Nature Comms pending major revisions

Response:

We agree with the importance of further demonstrating the physical realism of the model predictions and temperature-dependence, which is crucial to many process-oriented applications of our model. First, we would like to clarify that the compiled solubility data represent solubility limits, and thus are already at the precipitation points of these solutions. We have clarified this point in the **third paragraph of Section 1.1**.

The physical realism of the predicted gradients are now included for several specific solutions in **Figure 4** (prev. Appendix B, has been moved into main text) and **Supporting Information Sections S2 and S3**. Within these solutions we analyzed, we cover a range of solute and solvent classes. We evaluated model performance in several hydrogen-bonding solvents, including risperidone (**Figure 4a**), anthracene (**Figure S1c**), and anthraquinone (**Figure S1f**) in isopropyl alcohol, and N-acetylglycine in methanol (**Figure S2b**). In each of these cases, we observe that FastSolv extremely accurately captured the temperature-dependence of solubility, and in all cases except anthraquinone in isopropyl alcohol, we observe extremely accurate solubility predictions. In these highly polar solvents, we would expect non-ideal solution behavior, particularly since the solubility limit is at the precipitation point. We also evaluated predictions of L-proline solubility in highly non-polar solvents, heptane and hexane (**Figure 4b**), where the model demonstrates highly accurate predictions and gradients. Importantly, the model was also able to distinguish between solubility in these similar solvents, while the Vermeire et al. model is not able to (**Figure 4b**). Commentary on the model performance in these different case studies is also included in the **fourth, fifth, and sixth paragraphs of Section 1.2**.

Additionally, to further analyze performance of the model in predicting temperature-dependence, we plotted the distribution of the model prediction errors on the SolProp test set as a function of the temperature (below). We observe that both the fastprop-based and chemprop-based models exhibit uniform error distributions across temperature, further indicating the physical realism of temperature-dependence that the model captures.

We do note though that fastsolv predictions in the high temperature range (solutions above 350 K) will not continually increase with increasing temperature. This was a deliberate design choice given the nature of available experimental data. For solutions of organic solvents, solubility values above 350 K tend to be unreliable, since many organic solvents boil near this limit: ethanol (351.5 K), benzene (353.2 K), acetonitrile (355.1 K), methyl ethyl ketone (352.8 K), hexane (341.8 K), tetrahydrofuran (339.1 K), ethyl acetate (350.2 K). Thus, for the fastprop-based model packaged in fastsolv, we don't model temperature-dependence above 350 K, since the predictions would not account for evaporation or volatility. Instead, the model outputs the same prediction for any temperature above 350 K. We have included this limitation of the model and our rationale in **Supporting Information Section S1**. We believe that any organic solubility model should share this limitation, unless they explicitly include the physics of solvent (and solute) evaporation in their model.

Reviewer #2 (Remarks on code availability):

The GitHub repository associated with the paper is well-structured, with clear organization of files and directories. It provides a detailed README file that includes comprehensive instructions for installing and running the model, making it accessible for users in the community

to reproduce the results.

- However, one issue encountered was with the web-based version of the fastsolv predictor, fastsolv_web, hosted at fastsolv.mit.edu. Attempting to access this resource consistently raised an internal server error. This may hinder users who wish to explore the tool online, especially those without the expertise or resources to run the application locally. Addressing this issue would greatly enhance the accessibility and usability of the tool for the broader community.

Overall, while the repository itself is a valuable resource, ensuring that the web-based version is functional would further support the reproducibility and practical application of the work.

Response:

We appreciate the reviewer's effort to review our GitHub repository and their discovery of the bug on the fastsolv website. The website had a bug wherein submitting an empty form would crash the website, which we have now addressed. We plan to further improve both the functionality and appearance of the website.

References

1. Loftsson, T. (2010). Aqueous solubility and true solutions. *Die Pharmazie-An International Journal of Pharmaceutical Sciences*, 65(6), 404-407.
2. Andersson, S. B., Alvebratt, C., Bevernage, J., Bonneau, D., da Costa Mathews, C., Dattani, R., ... & Bergström, C. A. (2016). Interlaboratory validation of small-scale solubility and dissolution measurements of poorly water-soluble drugs. *Journal of Pharmaceutical Sciences*, 105(9), 2864-2872.
3. Pathak, Y., Mehta, S., & Priyakumar, U. D. (2021). Learning atomic interactions through solvation free energy prediction using graph neural networks. *Journal of Chemical Information and Modeling*, 61(2), 689-698.
4. Ross, J., Belgodere, B., Chenthamarakshan, V., Padhi, I., Mroueh, Y., & Das, P. (2022). Large-scale chemical language representations capture molecular structure and properties. *Nature Machine Intelligence*, 4(12), 1256-1264.
5. Chithrananda, S., Grand, G., & Ramsundar, B. (2020). ChemBERTa: large-scale self-supervised pretraining for molecular property prediction. *arXiv preprint arXiv:2010.09885*.
6. Cepeda, E. A., & Diaz, M. (1996). Solubility of anthracene and anthraquinone in acetonitrile, methyl ethyl ketone, isopropyl alcohol and their mixtures. *Fluid phase equilibria*, 121(1-2), 267-272.
7. Meng, J., Chen, P., Wahib, M., Yang, M., Zheng, L., Wei, Y., ... & Liu, W. (2022). Boosting the predictive performance with aqueous solubility dataset curation. *Scientific Data*, 9(1), 71.
8. Llompert, P., Minoletti, C., Baybekov, S., Horvath, D., Marcou, G., & Varnek, A. (2024). Will we ever be able to accurately predict solubility?. *Scientific Data*, 11(1), 303.